# Oxidative Stress and Its Modulation by Ladostigil Alter the Expression of Abundant Long Non-Coding RNAs in SH-SY5Y Cells

**DOI:** 10.3390/ncrna8060072

**Published:** 2022-10-25

**Authors:** Keren Zohar, Eliran Giladi, Tsiona Eliyahu, Michal Linial

**Affiliations:** Department of Biological Chemistry, The Institute of Life Sciences, The Hebrew University of Jerusalem, Jerusalem 91904, Israel

**Keywords:** neurodegenerative disease, RNA-seq, UPR, Nrf2 signaling, MALAT1, ceRNA

## Abstract

Neurodegenerative disorders, brain injury, and the decline in cognitive function with aging are accompanied by a reduced capacity of cells in the brain to cope with oxidative stress and inflammation. In this study, we focused on the response to oxidative stress in SH-SY5Y, a human neuroblastoma cell line. We monitored the viability of the cells in the presence of oxidative stress. Such stress was induced by hydrogen peroxide or by Sin1 (3-morpholinosydnonimine) that generates reactive oxygen and nitrogen species (ROS and RNS). Both stressors caused significant cell death. Our results from the RNA-seq experiments show that SH-SY5Y cells treated with Sin1 for 24 h resulted in 94 differently expressed long non-coding RNAs (lncRNAs), including many abundant ones. Among the abundant lncRNAs that were upregulated by exposing the cells to Sin1 were those implicated in redox homeostasis, energy metabolism, and neurodegenerative diseases (e.g., MALAT1, MIAT, GABPB1-AS1, NEAT1, MIAT, GABPB1-AS1, and HAND2-AS1). Another group of abundant lncRNAs that were significantly altered under oxidative stress included cancer-related SNHG family members. We tested the impact of ladostigil, a bifunctional reagent with antioxidant and anti-inflammatory properties, on the lncRNA expression levels. Ladostigil was previously shown to enhance learning and memory in the brains of elderly rats. In SH-SY5Y cells, several lncRNAs involved in transcription regulation and the chromatin structure were significantly induced by ladostigil. We anticipate that these poorly studied lncRNAs may act as enhancers (eRNA), regulating transcription and splicing, and in competition for miRNA binding (ceRNA). We found that the induction of abundant lncRNAs, such as MALAT1, NEAT-1, MIAT, and SHNG12, by the Sin1 oxidative stress paradigm specifies only the undifferentiated cell state. We conclude that a global alteration in the lncRNA profiles upon stress in SH-SY5Y may shift cell homeostasis and is an attractive in vitro system to characterize drugs that impact the redox state of the cells and their viability.

## 1. Introduction

A decline in cognitive function occurs along with brain aging. In humans and rodents, it includes structural changes and extensive neuronal cell death [1]. The young brain successfully copes with stress (e.g., oxidative stress), a capacity that drops in the aging brain [2,3]. Failure in the stress response leads to pathological processes, as documented in brain injury, ischemia and reperfusion (I/R), as well as neurodegenerative diseases [4,5].

Gradual changes in the gene expression profiles occur throughout the progression of normal aging and neurodegenerative diseases [6,7]. In previous studies, ladostigil, a bifunctional reagent with antioxidant and anti-inflammatory activities, was shown as a novel neuroprotective drug [8]. Old rats that were chronically treated with ladostigil showed improved performance in memory and learning tasks relative to untreated old rats [9,10,11]. Nevertheless, the direct effect of oxidative stress at a cellular level is masked by the communication and feedback circuits of different cell types, including the microglia, astrocytes, and endothelial cells with neurons. To overcome this difficulty, the antioxidant function of ladostigil was studied using a neuroblastoma cell line [8,12]. Upon oxidative stress, an imbalance in mitochondrial redox was monitored, which was attenuated by ladostigil [12].

Emerging evidence suggests the involvement of long non-coding RNAs (lncRNAs) in neurodevelopmental, metabolic, and inflammatory diseases [13]. Furthermore, specific lncRNAs are implicated in all major neurodegenerative diseases, including Alzheimer’s disease (AD) [14,15]. Specifically, the highly expressed lncRNAs BACE1-AS, 51A, and BC200 were implicated in the production and accumulation of Aβ and were implicated in AD pathology in mice with AD pathology; SOX21-AS1 expression determines the level of neuronal oxidative stress and suppresses neuronal apoptosis [16]. Other lncRNAs have been linked to Parkinson’s disease (PD) [17,18]. For example, HOTAIR induces the apoptosis of dopaminergic neurons and MALAT1 (also called NEAT2) is associated with α-synuclein expression and PD pathogenesis [19]. The role of lncRNAs in governing the cellular oxidative levels was investigated beyond the context of neurodegenerative diseases and brain injury. Examples include oxidative stress that impacts pancreatic functioning [20], cardiac myocytes [21], lung epithelia [16], and more. Multiple layers of regulation are involved in lncRNA modulation, including chromatin remodeling [20], the regulation of gene expression enhancers [22], competition for miRNA binding sites on mRNAs [23], the recruitment of transcription factors to promoters, alterations in splicing, interference with expression as antisense transcripts, molecular decoys, and more [24,25]. Moreover, the involvement of circRNAs in manipulating oxidative stress has been established in cell systems and in the pathogenesis of diseases of the CNS [26]. Studies have shown a correlation between oxidative stress and lncRNA expression [27]. However, the mechanism by which lncRNAs can change in response to drugs that alter cellular homeostasis is mostly unknown.

In this study, we studied SH-SY5Y cells, a neuroblastoma-derived cell line that carries some dopaminergic neuronal features [28]. We tested the expression profiles of the cells under varying conditions by RNA-seq [12]. We exposed the SH-SY5Y cells to oxidative stress and monitored their viability and gene expression changes [12]. We applied conditions that resembled abrupt (by H_2_O_2_) and prolonged (by Sin1; 3-morpholinosydnonimine) oxidative stress and monitored the beneficial effect of ladostigil on the viability of SH-SY5Y cells under stress conditions. We focused on the mild upregulation of abundant lncRNAs following treatment with Sin1 and determined the impact of ladostigil under oxidative stress conditions on lncRNA expression. Our investigation of the gene expression levels along with the attenuation of cell death by ladostigil suggests the beneficial effect of lncRNAs on cell viability while coping with oxidative stress.

## 2. Results

### 2.1. Ladostigil Suppresses Cell Death

Figure 1A shows the results of flow cytometry to determine the fraction of necrotic cells (each experiment covered 50,000 cells). Propidium iodide (PI) was used as a marker for permeable membranes of dead cells. After 5 h of exposure to 40 μM H_2_O_2_, we saw a significant increase in PI-positive cells. In the presence of ladostigil, the fraction of PI-positive cells was attenuated (8.0% to 5.4%; *p*-value = 0.02). The attenuation in cell death was also detected 24 h later (Appendix A). After treatment with ladostigil, the fraction of PI-positive cells in cells exposed to 40 μM and 80 μM H_2_O_2_ was reduced (Appendix A). Therefore, we conclude that ladostigil protects cells against non-reversible cell death induced by acute and short-lived stress of H_2_O_2_.

### 2.2. Apoptotic Signal in SH-SY5Y Cells Exposed to Sin1

To better mimic a prolonged and steady increase in oxidation, we exposed the cells to Sin1 at varying concentrations. Sin1 in living cells acts as a peroxynitrite donor (ONOO^−^) and generates nitric oxide (NO) and superoxide (O_2_^−^) [29,30]. In addition to the effect of Sin1 on the redox state of cells, it was reported as an inhibitor of high-affinity choline transporters and trafficking in SH-SY5Y cells [31]. We monitored cell viability (using MTT) by increasing the levels of Sin1 (0 to 350 mM). In parallel, we performed the same experiment but with the preincubation of ladostigil (5.4 μM, 2 h prior to Sin1 exposure). Calculating the IC50 for each of the graphs resulted in the IC50 for Sin1 being 247.0 μM, while the ICD50 of ladostigil in the presence of Sin1 changed to 350.4 μM. These results allow the beneficial effect of ladostigil in coping with Sin1-induced cellular stress to be quantified (Figure 1B). The effect of prolonged oxidative stress produced by high levels of Sin1 on cell viability was assessed by flow cytometry using PI (Figure 1C). Cell viability was tested at 5 h upon increasing the concentration of Sin1. Under these harsh conditions, we also stained SH-SY5Y cells with Annexin-V, a sensitive marker for early apoptosis. Staining cells with Annexin-V showed that the fraction of apoptotic cells was insensitive to increased levels of Sin1 (Figure 1D). We concluded that, under a high concentration of Sin1, cell death was increased. However, the fraction that underwent apoptosis remained stable (~10%), supporting an increase in necrotic cells, rather than apoptosis. Recall that, at later time points, separating cells by PI and Annexin-V positives is no longer valid, as the membranes of the cells become fully permeable (discussed in [32]).

### 2.3. LncRNA Expression Levels in SH-SY5Y Cells

Sin1 at a low concentration (50 μM) produces mild but long-lasting stress due to the ongoing accumulation of nitric oxide (NO) and superoxide (O_2_^−^). We focused on the ncRNA levels using RNA-seq (RNA length cutoff at >200 nt) 24 h after treatment. Figure 2A shows that the cells’ transcriptome comprised 70% mRNAs of coding genes (13,626 coding mRNAs out of a total of 19,475 transcripts), with the rest of the transcripts being partitioned among many ncRNA biotypes. The dominant class was lncRNA (long ncRNA, 63.2% of all ncRNAs), followed by pseudogenes (28.8%). However, the expression levels measured by TMM for most of these transcripts were very low (<4 TMM for biological triplicates, see the Section 4). By considering only transcripts that satisfied this threshold, we focused on 780 ncRNAs expressing transcripts that accounted for 6.7% of all transcripts (total of 11,588, Figure 2B). The rest of the analysis will consider only these ncRNAs (Appendix A). A subdivision according to biotypes confirmed that most ncRNA transcripts belonged to the lncRNA group (73%), followed by the unprocessed pseudogenes group (19%, Figure 2B).

Figure 2C shows a sorted list of high-expressing lncRNAs in SH-SY5Y cells (above TMM of 100). A manual classification of these highly expressed transcripts into groups revealed that about 16% of these transcripts belonged to mitochondrial ncRNAs (e.g., MT-RNR1 and MT-RNR2 for 12S and 16S rRNAs of the mitochondrial ribosome, respectively), and an additional 10% were transcripts involved in X-chromosomal inhibition and chromatin compaction (e.g., XIST, FTX, and JPX, collectively termed XCI, Figure 2C). Several of the other abundant lncRNAs (Figure 2C) were implicated in the regulation of oxidative stress, particularly in the context of the CNS and neurons (e.g., MALAT1, GAS5, and NEAT1) [33].

### 2.4. Chronic Oxidative Stress Upregulated a Collection of LncRNAs

Studies have shown a dependency between acute oxidative stress and the upregulation of lncRNA expression [27,34]. We tested the profile of the cells’ ncRNA levels 24 h after the Sin1 stimulus. Figure 3A shows the number of differentially expressed (DE) ncRNA transcripts partitioned into ncRNA types. We found that the strongest effect induced by Sin1 was the upregulation of ncRNAs. The fraction of ncRNAs among DE downregulated genes was only 5.6%, while among the upregulated transcripts, it accounted for 17.1% (*p*-value = 0.012). Moreover, among the downregulated genes, the ratio of lncRNAs to all other molecular ncRNA biotypes was 2.1, but it was 5.1 for the upregulated genes (Figure 3A). We conclude that the fraction of the expressed ncRNAs, and specifically the lncRNA class, significantly increased under Sin1-induced stress. While many coding genes were induced by Sin1 [12], the ncRNA expression fold change was modest (bounded by <3-fold, Figure 3B).

Table 1 shows the list of lncRNAs following Sin1 treatment (listed genes with >0.5% of total expression, see the Section 4). All 18 Sin1-upregulated ncRNAs (Table 1) were statistically significant (*p*-value FDR < 1.0 × 10^−3^; Appendix A). The highest-expressed lncRNA was MALAT1 (Figure 2C), which was also upregulated by Sin1 (Table 1, *p*-value FDR < 2.5 × 10^−4^).

Table 1 presents the Sin1-induced lncRNAs that met the threshold of absolute expression and that were statistically significant. Among them were NEAT1 (upregulated 1.49-fold, FDR *p*-value 3.5 × 10^−5^) and HAND2-AS1 (upregulated 1.33-fold, FDR *p*-value 6.7 × 10^−25^), both implicated in oxidative stress modulation. Notably, among the Sin1-upregulated genes, many were annotated as antisense (21% of all transcripts upregulated by Sin1; Appendix A). Table 1 also lists lncRNAs that were downregulated by Sin1 (five transcripts, Appendix A). Note that, in comparison with the upregulated lncRNAs, the absolute expression levels of these genes were low (<40 TMM).

### 2.5. Cancer-Related SNHG Family Members Are Differentially Expressed by Sin1

SNHGs (small nucleolar RNA host genes) belong to a large family of lncRNAs whose expression greatly fluctuates among cell types and conditions [35]. We observed 17 SNHGs expressed transcripts, which accounted for 3.3% of all highly expressed lncRNAs (Appendix A, 780 genes).

Figure 4A presents the expression levels (log_10_TMM) of the identified SNHGs. Four of the SNHGs (SNHG5, SNHG14, SNHG16, and SNHG29) were highly expressed, with >100 TMM each. The fold change in expression following Sin1 treatment was small to modest. SNHG3, SNHG4, SNHG12, and SNHG15 were significantly downregulated by Sin1. SNHG29 and SNHG5, which were highly expressed lncRNAs (TMM 328.3 and 131.4, respectively), were slightly upregulated by 10.1% and 8.1%, respectively (FDR *p*-value of 1.6 × 10^−4^ and 0.029; Figure 4, Appendix A). Our observations are consistent with the proposed role of SNHGs in miRNA competition and as molecular decoys in SH-SY5Y [36].

### 2.6. Impact of Ladostigil on ncRNA Profiles

To test the potential of ladostigil to cope with prolonged mild stress by Sin1 (50 μM, 24 h), we applied ladostigil (5.4 mM) and measured the effect on gene expression 24 h after treatment. Under such mild conditions of oxidation, the level of cell death was minimal (below 10%) [12]. As a control, we incubated cells with ladostigil for 2 h, which had negligible effects on cell viability (24 h post-incubation, Appendix A).

Figure 5A shows four of the most significantly upregulated lncRNAs by Sin1 (*p*-value FDR ranged from 1 × 10^−23^ to 1 × 10^−51^), but were unchanged in the presence of ladostigil (24 h). Among these transcripts were TSPEAR-AS1 and TSPEAR-AS2. TRPEAR is a low-expressing protein that participates in the regulation of the Notch signaling pathway. While information on antisense transcripts is scarce, samples of kidney renal clear cell carcinoma with higher TSPEAR-AS1 expression showed poor survival when compared with patients with low expression (hazard ratio of 1.95; *p*-value of 8.8 × 10^−6^) [37]. The other examples, AC106881.1 (UNC5C antisense RNA 1; Figure 5A) and BX005019.1, are exclusively expressed in brain tissues. BX005019.1 is located within the MIR137 host gene, a main player in the differentiation and maturation of the nervous system [38].

We tested whether ladostigil reduces the damage induced by oxidative stress via lncRNA expression. To this end, we considered the expression profile following Sin1 as a reference, and applied a strict statistical threshold. Only 8 genes out of 19,475 annotated genes were identified as significantly induced by ladostigil, with 4 coding genes and 4 lncRNAs (Figure 5B). RPARP-AS1 (also called C10orf95 Antisense RNA 1) overlapped with the production of a short (213 aa) protein of unknown function (Figure 5C). Possible regulation of the splicing machinery was proposed for AC105339.2, which overlapped with the FSD2 transcript (fibronectin type III and SPRY domain containing 2; also called A1L4K1). SNHG21 and SCARNA15 (Figure 5C) were two other overlapping genes that were implicated in the redox homeostasis of cancer cells and stress conditions [39].

We further inspected the genomic organization of these four ncRNAs upregulated by ladostigil in cells exposed to Sin1 (24 h). We used GeneHancer (GH) [40] to test the likelihood of the ladostigil-upregulated ncRNAs acting as enhancers (eRNAs). GeneHancer integrates information from numerous independent databases.

Table 2 presents the prediction confidence for the four ladostigil-upregulated ncRNAs as enhancers in view of their potential gene target regulation. Elite enhancers were defined by high confidence scores for enhancers according to multiple independent resources. All of the genes listed were classified as elite enhancers. Note that only 7% of the >100k analyzed genes and transcripts in the GH database were marked as elite enhancers. GH double elite was assigned to two of the four transcripts. Under stress, ladostigil induction of lncRNAs may impact global processes, as reflected by eRNAs that act as enhancers.

It is an accepted view that each lncRNA molecule is involved in multiple cellular functions. For example, we propose a function for AC105339.2 as an enhancer (eRNA, Table 2) and also as an antisense (Figure 5). Moreover, the human RNA interactome has a wealth of experimental evidence for protein, miRNA, and lncRNA interactions [41]. The notion of multi-function activities for lncRNAs was validated for several abundant ones. Both GAS5 and the mitochondrial pseudogene MTND4P12 were among the most abundant lncRNAs in SH-SY5Y (Figure 2C), and both were significantly upregulated in the presence of ladostigil (Appendix A). GAS5 is implicated in human cancer as a tumor suppressor that is modulated by miRNAs. In SH-SY5Y cells exposed to Sin-1, the level of GAS5 is already high (TMM = 188) and is slightly upregulated by ladostigil (*p*-value 5 × 10^−5^). We report on 45 MBSs on GAS5 (mapped to 35 unique miRNAs, Appendix A) using a stringent seed-matching scheme [42]. Interestingly, the GAS5 MBSs contained several highly expressed miRNAs, such as miR-10b (21.8% of all expressed miRNAs in undifferentiated SH-SY5Y), miR-21, miR-148a, miR-26a, miR-222, and more. We expect that the increase in GAS5 transcripts by ladostigil will directly bind miRNAs and consequently impact cell survival through ceRNA activity (Appendix A). A similar activity of ceRNA can be performed by MTND4P12 (MT-ND4 pseudogene 12), which has MBSs for 10 different miRNAs, among which miR-140 and miR-24 are relatively abundant in SH-SY5Y cells.

### 2.7. A Shift in Abundant ncRNAs upon Stress Is Cell-State-Dependent

SH-SY5Y cells can undergo in vitro differentiation with retinoic acid (RA), and as a result, develop a predominantly mature dopaminergic-like neuronal phenotype [43]. We performed semi-quantitative RT-PCR to test the impact of gene expression change for several of the major ncRNAs that were implicated in the response to oxidation stress according to the cell state (undifferentiated vs. RA differentiated). We assessed the effect on gene expression following the exposure of cells to Sin1 only and to Sin1 in the presence of ladostigil (5.4 μM; Figure 6). The results from the gel analysis of the amplicons of the lncRNAs indicate that the differentiated cells were signified by reduced expressions of MALAT-1 and MIAT, but not NETA-1. In accordance with the results from the RNA-seq analysis, the expression levels of MALAT-1, MIAT, and NEAT-1 were induced by Sin1, and ladostigil partially reverted the elevated expression of these major lncRNAs (no change was observed in β-actin as an internal control; Figure 6A). The maximal upregulation in expression was measured in MIAT (lncRNA myocardial infarction-associated transcript). The MIAT was shown to maintain human lens epithelial cells (HLECs), where its expression is critical to the viability, proliferation, and migration of the cells in response to oxidative stress [27]. Overall, Sin1 induced many of the abundant ncRNAs in the undifferentiated cells, while ladostigil attenuated the effect. In contrast, the effect of Sin1 was quite modest and no induction was observed in undifferentiated cells.

## 3. Discussion

In this work, we examined undifferentiated SH-SY5Y cells under oxidative stress conditions by inspecting the lncRNA expression profiles [44,45,46]. In brain injury, ischemia, neuroinflammation, and other pathophysiological processes, changes in the amount and composition of ncRNAs (e.g., miRNAs and lncRNAs) are frequently observed [47,48]. Several discoveries regarding ncRNAs emerged from the analysis of SH-SY5Y under continuous oxidative stress. Firstly, although ncRNAs accounted for 30% of all transcripts detected, many were lowly expressed (only 6.7% with TMM > 4, Figure 2), reflecting the high sensitivity of the sequencing methodology [49]. Secondly, following sin1, most lncRNAs were upregulated, with only a few being downregulated (Figure 3, Fisher exact test, *p*-value < 1 × 10^−5^). Finally, the absolute fold change (FC) of the Sin1-upregulated lncRNAs was low (Table 1). We argue that, even with such a low increase or decrease in the overall expression (Figure 3B bounded by 3.0-fold), the dysregulation of abundant lncRNAs substantially impacts cells’ states.

Despite the growing interest in using lncRNAs for prognosis [50], knowledge of lncRNA binding specificity is limited, with only a few highly expressed lncRNAs having been studied in depth. Even under the mild oxidation stimulation that was introduced to undifferentiated SH-SY5Y cells (50 μM Sin1, 24 h), tens of lncRNAs were up-regulated, a few of them belonging to the highly expressed lncRNAs. Many of the highly expressed lncRNAs have been linked to neurodegenerative, inflammatory, metabolic, and systemic diseases [33,51]. It is likely that the same lncRNAs may act on multiple layers of regulation, such as chromatin, transcription, biological machines (e.g., spliceosomes), recruiting or stabilizing proteins, and competing for binding sites [52]. Inspecting the genomic location supports the notion of multiple functions that can be carried out by a specific lncRNA [53]. For example, GABPB1-AS1 (Table 1) suppresses the transcription of GABPB1, a gene product that acts upon damaged membranes via peroxidases. As a result, in both health and disease, it regulates cellular antioxidant capacity and cell viability [15]. Several of the most abundant lncRNAs that were affected by changing conditions (e.g., MALAT1, XIST, GAS5, and NEAT1) [24] have been proposed to act via multiple modes of action [52]. NEAT1 lncRNA, in particular, was initially linked to chromatin organization but has since been shown to play a role in regulating oxidative stress, pain control, and inflammation [54,55]. In neurons, NEAT1 could reverse the damage caused by superoxide and counteract the H_2_O_2_-induced neuronal damage. The dysregulation of HAND2-AS1 can alter energy metabolism at the organismal level [56]. MALAT1, the most abundant lncRNA in SH-SY5Y, is involved in a variety of cell biology processes, including autophagy and ischemia-induced apoptosis [57,58]. MALAT1 and NEAT1 play a neuroprotective role in AD and PD, respectively [59,60]. MIAT, NEAT1, and MALAT1 are coordinately changed in schizophrenia [61]. Other lncRNAs have been linked to autism, amyotrophic lateral sclerosis (ALS), and other diseases.

The direct role of miRNAs in maintaining SH-SY5Y cell homeostasis has been extensively studied [62]. Several abundant ncRNAs play an active role in such sponging mechanisms. For example, MALAT1 promotes oxygen–glucose deprivation/reoxygenation (OGD/R)-induced neuronal injury through miRNA competition [59,60]. Similarly, NORAD promotes cell survival by sponging miRNAs [23]. In such cases, even a small change in the accessibility of miRNAs alters the ratio of free to bound cellular miRNAs, leading to a revised cell state by the available miRNA pool [63].

A special case concerns the SNHG genes (Figure 4) that were originally studied in cancer and are characterized by a rich repertoire of miRNA binding sites (MBS). As a result, several of the SNHGs may collaboratively titrate out miRNAs and thus release their bound targets, a mechanism known as ceRNA [25,64]. Of these, SNHG12 was significantly downregulated in the presence of Sin1 in undifferentiated SY-SY5Y cells. In these cells, SNHG12 participates in the unfolded protein response (UPR), apoptosis, inflammation, and oxidative stress [65]. It has been proposed that several of the SNHG members carry anti-inflammatory roles under oxygen–glucose deprivation/reoxygenation (OGD/R) conditions by targeting specific miRNAs [33,66]. As a general theme, lncRNAs, such as SHNGs, act via the sponging of specific miRNAs (e.g., [67,68]).

Among the abundant neuronal lncRNAs that were altered by Sin1 treatment, MALAT-1, MIAT, NEAT1, and MIR137HG (Table 1) contain miR-106a MBSs [69]. This also applies to members of the SNHG lncRNAs (Appendix A). Specifically, we identified 17 members of the SNHGs that were expressed in SH-SY5Y (Figure 4). Note that MBSs for miR-106a were found in nine of them (SNHG1, SNHG3, SNHG5-7, SNHG10-12, and SNHG14, Appendix A). Many of these targets were confirmed by miR-CLIP with miR-106a as bait [70]. An inverse link was established between the level of available miR-106a and Treg anti-inflammatory response (e.g., allergy, inflammatory bowel disease (IBD), and LPS-induced macrophages) [71]. In SH-SY5Y, we anticipate that the collaborative sponging of miR-106a by abundant lncRNAs, including SNHGs, underlies the beneficial impact of ladostigil.

Interestingly, among the ncRNA-induced genes following Sin1 is MIR137HG (Table 1), a precursor of hsa-miR-137 and hsa-miR-2682, two highly expressed miRNAs of the CNS and especially the brain cortex. In genome-wide association studies, the MIR137HG gene has been linked to schizophrenia and other psychiatric disorders [72,73]. The function of has-miR-137 and has-miR-2682 in cell proliferation was reported in the context of cancer, where SNHG3 acts as a direct sponge of miR-2682 [74]. Thus, it is likely that the suppression of SNHG3 (Figure 4) and the induction of MIR137HG (1.5 fold, *p*-value 4.5 × 10^−23^, Table 1) together increased the accessibility of miR-137 and miR-2682.

An attractive hypothesis for the involvement of abundant ncRNAs under long-lasting stress conditions concerns regulation by the antioxidation response. We recently showed that oxidative stress in SH-SY5Y undifferentiated cells led to an alteration in ER stress and the redox state [12]. Nrf2 (gene name: NFE2L2), a master transcription factor (TF) of redox homeostasis, is upregulated by Sin1, whereas ladostigil inhibits NFE2L2 expression [12]. Furthermore, Nrf2 has been identified as a regulatory hub for lncRNAs [75] and miRNAs [76,77]. The capacity of ladostigil to improve cell viability (Figure 1) could reflect a cross-talk of MALAT1 and Nrf2 in cytotoxicity. We identified MALAT1, the highest expressed lncRNA in undifferentiated SH-SY5Y cells, to be further upregulated by Sin1 (1.24-fold). MALAT1 has been shown to inhibit apoptosis via Nrf2 activation. This is mediated via the suppression of lipid peroxidation and damaged DNA [60].

In this work, we focused on undifferentiated SH-SY5Y cells exposed to abrupt (by H_2_O_2_) and long-term oxidative stress (by Sin1). However, exposing SH-SY5Y cells to various combinations of growth factors and morphogens (e.g., RA) drives their differentiation toward mature dopaminergic-related neurons [78,79]. After exposing SH-SY5Y to RA for several days, profound changes in mitochondrial metabolism were observed [80]. While differentiated SH-SY5Y cells maintained the same mitochondrial number, they were distinguished by elevated membrane potentials. Overall, the differentiated cells were more resistant to cytotoxicity and mitochondrial dysfunction. In accordance with these results, we show that RA-differentiated SH-SY5Y coped with Sin1 where no induction of abundant ncRNAs was recorded (Figure 6). Importantly, alternative differentiation protocols that were presented (e.g., the two-step RA and BDNF combination) drove the cells to become more vulnerable to energetic stress, despite the increased ATP production [28]. We anticipate that inspecting specific changes in the lncRNA profile is useful for the study of functional neurons in health and disease [51].

In summary, we present evidence for reduced cytotoxicity in undifferentiated SH-SY5Y cells that were treated by ladostigil. We found that ladostigil shifted the IC50 for cell viability of Sin1 from ~250 μM to ~350 μM, presumably by its antioxidative activity. There were numerous mechanisms by which lncRNAs responded to oxidative stress, including the regulation of transcription by enhancers, competing with miRNAs, modulating transcription factors, and molecular decoys. Among the hundreds of lncRNAs identified in the cells (Appendix A), tens of abundant ones were moderately upregulated. We show that RA-differentiated cells were quite robust to Sin1 and ladostigil. We anticipate that the oxidative stress-driven shifts in the lncRNA expression levels participated in the cells’ responses to external stress, which distinguished their differentiated state. We propose that even a modest dysregulation in the lncRNA levels may affect cell homeostasis. We propose lncRNAs as cellular indicators for assessing the effects of novel drugs (e.g., ladostigil) under chronic stress conditions that characterize the aging brain and neurodegenerative diseases.

## 4. Materials and Methods

### 4.1. Materials

All reagents were purchased from Sigma Aldrich (Burlington, MA, USA) unless otherwise stated. Ladostigil, (6-(N-ethyl, N-methyl) carbamyloxy)-N propargyl-1(R)-aminoindan tartrate, was a gift from Spero BioPharma (Cambridge, MA, USA). Sin1 (3-(4-5 Morpholinyl) sydnone imine hydrochloride) was obtained from Sigma-Aldrich (Cat-M5793). Media products MEM and F12 (ratio 1:1), heat-inactivated fetal calf serum (10% FCS, Cat. C8056, Sigma-Aldrich), and L-Alanyl-L Glutamine were obtained from Biological Industries IMBH (Beit-Haemek, Israel). All tissue culture materials were purchased from IMBH, (Beit-Haemek, Israel). H_2_O_2_ (30%, Cat. H1009, Sigma-Aldrich, Rehovot, Israel) was diluted in cold water and the solutions were used within 2 h. Cell viability was determined using MTT (3-(4,5-dimethylthiazol-2-yl)-2,5-diphenyltetrazolium bromide; Cat M5655, Sigma-Aldrich, Rehovot, Israel).

### 4.2. SH-SY5Y Cell Culture

Human neuroblastoma-derived SH-SY5Y cells derived from the original SK-N-SH clone were obtained from the ATCC (American Type Culture Collection, Manassas, VA, USA) [78]. Cells were cultured in Minimum Essential Media (MEM and F12 ratio 1:1, 4.5 g/l glucose) with 10% fetal calf serum (FCS) and 1:10 L-Alanyl-L-Glutamine. Cells were incubated at 37 °C under a humidified atmosphere of 5% CO_2_. Ladostigil was added to the culture medium 2 h prior to the activation of oxidative stress, and analysis was performed. Cells were tested 24 h later (i.e., 26 h after the addition of ladostigil), unless otherwise stated.

SH-SY5Y cells were differentiated by retinoic acid (RA) according to the published protocol [81]. Briefly, 1.5 × 10^4^ cells/cm^2^ were seeded in medium supplemented with 10% fetal bovine serum (FBS, Cat. F7524, Sigma-Aldrich). After 24 h, the cells were introduced to a fresh medium with only 1% FBS, supplemented with 10 µM of RA (all-trans-RA, Cat. R625, Sigma-Aldrich), and maintained for an additional 7 days. On day 4, the same medium was replaced with fresh medium. On day 7, the cells were harvested and used for the preparation of the cDNA for the RT-PCR experiments.

### 4.3. Cell Viability Assay

SH-SY5Y cells were cultured at a density of 2 × 10^4^ cells per well in 96-well plates with 200 µL of medium. An MTT assay was conducted using colorimetric measurement for cell viability, with viable cells producing a dark blue formazan product [82]. The cells were treated in the absence or presence of ladostigil (at 5.4 mM and 54 mM). The drug is soluble in water. The MTT solution in phosphate-buffered saline (PBS, pH 7.2) was prepared as a working stock of 5 mg/mL. After 24 h, the culture medium was supplemented with 10 µL of concentrated MTT per well. Absorption was determined in an ELISA reader at *λ =* 535 nm, using absorption at *λ* = 635 nm as a baseline. Cell viability was expressed as the percentage of untreated cells. Each experimental condition was repeated 8 times. The results of the survival assays are presented as the mean ± SEM. When appropriate, *p*-values < 0.05 were calculated and considered statistically significant. We report on genes that met the FDR-adjusted *p*-value of ≤0.05 (q-value).

### 4.4. Flow Cytometry

SH-SY5Y cells were cultured in 6-well or 12-well plates to reach a 70–80% confluence level (37 °C and 5% CO_2_) before analysis by fluorescence-activated single-cell sorting (FACS). Propidium iodide (PI, Cat. R37108, Thermo-Fisher (Waltham, MA, USA) was used as a marker for dead cells. PI and Annexin V-FITC were purchased from MBL (MEBCYTO-Apoptosis Kit; Woburn, MA, USA). Early apoptosis was detected using conjugated Annexin V [83]. The fluorescence of Annexin-V indicates the presence of phosphatidylserine (PS) at the outer leaflet of the plasma membrane. FACS analysis was conducted with 50,000 cells and the fraction of dead cells stained with PI was determined. Cells were stained with Annexin V according to the manufacturer’s MBL protocol. Cells in the early apoptosis phase are defined as Annexin V-positive and PI-negative. Discriminative staining with PI and Annexin-V by FACS was discussed in [32,84].

### 4.5. RNA Sequencing

SH-SY5Y cells were pre-incubated for 2 h with ladostigil at a concentration of 5.4 μM. The cells were exposed to Sin1 (t = 0) and harvested at 10 h or 24 h following the addition of ladostigil. The total RNA was extracted using an RNeasy Plus Universal Mini Kit (QIAGEN, GmbH, Hilden, Germany) according to the manufacturer’s protocol. Briefly, a mini spin column was used and centrifuged for 15 s at ≥8000× *g* and room temperature. Samples with an RNA integrity number (RIN) of >9.0, as measured by an Agilent 2100 Bioanalyzer, were considered for further analysis. Total RNA samples (1 μg RNA) were enriched for mRNAs by the pull-down of poly(A) RNA and libraries were prepared using a KAPA Stranded mRNA-Seq Kit (QIAGEN, GmbH, Hilden, Germany) according to the manufacturer’s protocol. As the RNA-seq experiment was performed using a protocol for the removal of ribosomal RNAs, restricting molecule length to a minimum of >200 nucleotides, a sequencing depth to many of the lowly expressing transcripts could be achieved. RNA-seq libraries were sequenced using Illumina NextSeq 500 to generate 85 bp single-end reads. RNA-seq data files were deposited in ArrayExpress [85] under accession number E-MTAB-10450.

### 4.6. Reverse Transcription Polymerase–Chain Reaction (RT-PCR)

Total RNA was extracted with Trizol (Thermo-Fisher, Waltham, MA, USA)), and RT was performed using a Ready-To-Go first-strand synthesis kit (Amersham Pharmacia Biotech, Cat. 27925901, Uppsala, Sweden) according to the manufacturer’s instructions. RNA was reverse-transcribed into cDNA (1 μg) and used in the PCR reaction. The primers used for the PCR reactions are listed in Table 3. The PCR conditions consisted of denaturation at 95 °C for 2 min and 35 cycles (10 s at 95 °C, 15 s at 60 °C, and 15 min at 72 °C for extension). The amplicon from the β-actin was used as the internal control for normalization. The PCR products were separated on 1.5–2% agarose gel and stained with ethidium bromide, followed by densitometry measurement (using ImageJ).

### 4.7. Bioinformatic Analysis and Statistics

All next-generation sequencing (NGS) data underwent quality control using FastQC [86] and were processed using Trimmomatic [87]. All genomic loci were annotated using GENCODE version 37 and aligned to GRCh38 using a STAR aligner [88] and default parameters. All statistical tests were performed using R-base functions. Figures were generated using the R package ggplot2. The experiments contained a minimum of three biological replicates. The trimmed mean of M-values (TMM) was used for the normalization of the RNA read counts. A TMM value of > 4 was used as the lower cutoff value for the gene expression levels. For quantifying relative expression, the sum of TMM values from all lncRNA (> 4) was considered as 100% and only ncRNAs representing > 0.5% of that total TMM are reported. Differential expression analysis was performed using edgeR [89]. Unless otherwise stated, genes with a false discovery rate (FDR)-adjusted *p*-value of ≤ 0.05 and an absolute log fold-change (FC) above 0.3 were considered as significantly differentially expressed. Data on the miRNA–lncRNA–protein interactome were downloaded from the experimentally verified RAID v2.0 database [69].

## Figures and Tables

**Figure 1 ncrna-08-00072-f001:**
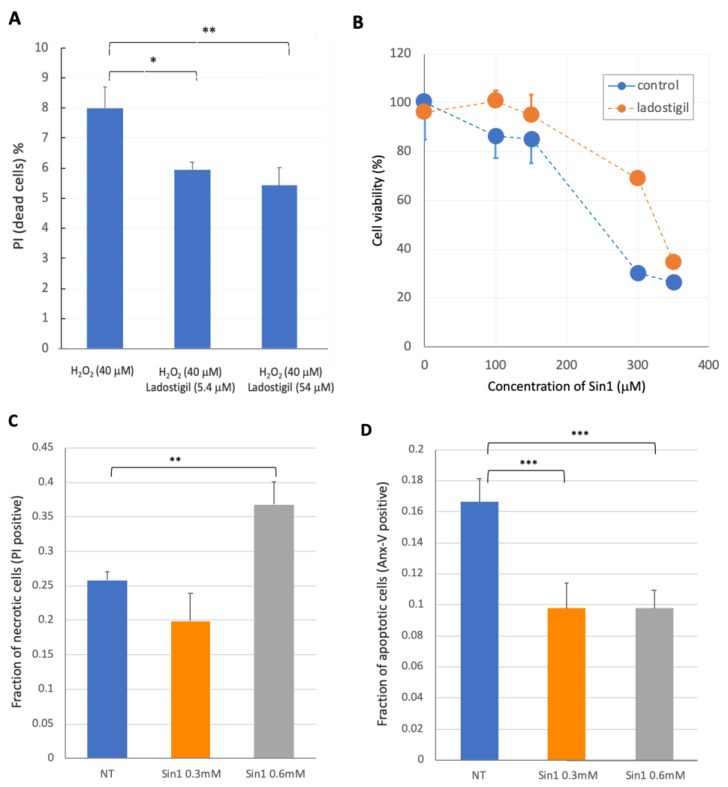
Flow cytometry analysis in cells under oxidative stress conditions in response to treatment with ladostigil. (**A**) Results from FACS analysis with 50k cells per experiment. The fraction of live and dead cells upon 40 μM H_2_O_2_ and ladostigil treatment was tested at 5 h following stimulus. Asterisks indicate the statistical significance of the results. The attenuation in the fraction of PI-positive cells by ladostigil (5.4 μM) was substantial (*p*-value = 0.02). (**B**) Cell viability upon increasing the concentration of Sin1 with and without the presence of ladostigil (5.4 μM, with mean and standard error; s.e.). Each data point is based on 8 replicates. (**C**) Results from 3 repeated experiments with mean and standard deviation (s.d.) from the flow cytometry run of SY-SY5Y cells treated with Sin1 for 5 h. All measured data were from FACS that was repeated in triplicate. The total FACS cells was considered as 100%, and the percentage of them stained with PI is indicated as a fraction. (**D**) Results from triplicate experiments with the mean and standard deviation (s.d.) from the flow cytometry run for Annexin-V of SY-SY5Y cells treated with Sin1 for 5 h. NT indicates cells non-treated with Sin1. Asterisks indicate the *t*-test statistical results with *p*-values of <0.005 (***), <0.01 (**), and <0.05 (*).

**Figure 2 ncrna-08-00072-f002:**
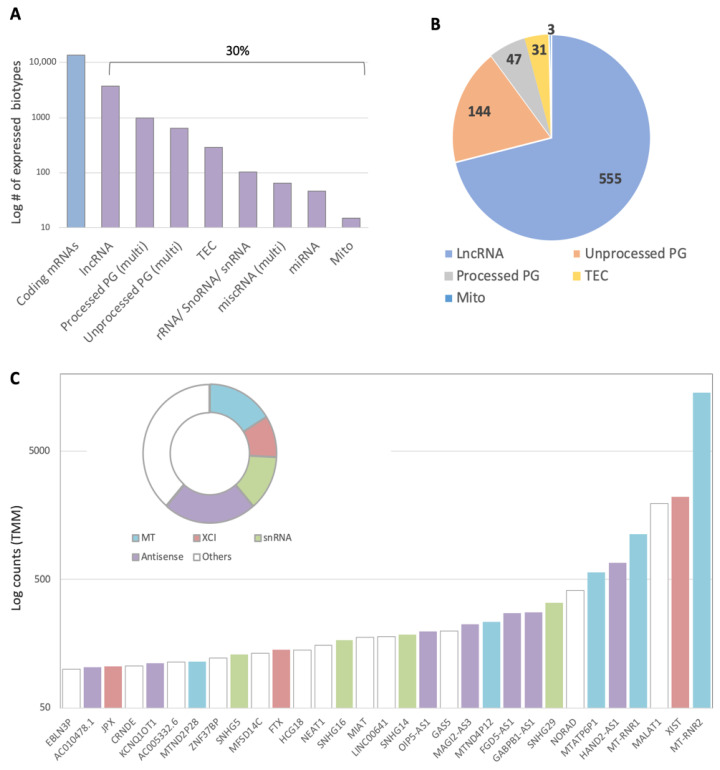
Partition of ncRNA biotypes in SH-SY5Y cells. (**A**) Partition of the RNA-seq results of untreated SH-SY5Y cells according to the major molecular biotypes. Altogether, there were 19,475 identified transcripts. Blue, coding genes and purple, ncRNAs portioned by biotype classes. Mito, mitochondrial transcript; TEC, to be experimentally confirmed; and PG, pseudogene. Combined related biotypes (e.g., transcribed unprocessed and unitary pseudogenes) are marked as multi. (**B**) Partition of 780 expressed ncRNA transcripts partitioned by major ncRNA classes. The ncRNAs were based on transcripts with an average TMM of >4. There were 11,588 aligned transcripts that met this threshold. (**C**) Top expressing ncRNAs (TMM > 100) colored by broad-sense functional annotations (inset). MT, mitochondria; snRNA, small nucleus RNA; and XCI, X-chromosome inactivation.

**Figure 3 ncrna-08-00072-f003:**
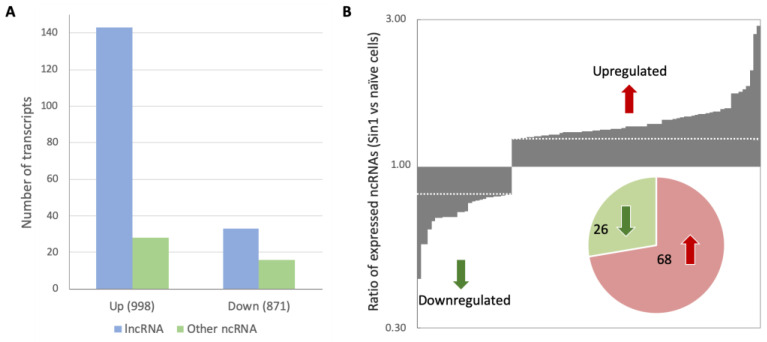
Differentially expressed (DE) ncRNAs following 24 h of exposure to Sin1 relative to untreated cells. (**A**) Transcripts associated with upregulation (Up) and downregulation (Down) relative to naïve, untreated cells. The number of genes associated with each expression trend is shown in parentheses. The rest of the genes (9719 genes) were unchanged. See the Materials and Methods for the expression trend definition. (**B**) Histogram showing the DE ncRNAs ratio for downregulated and upregulated transcripts. The dashed line marks the expression ratio (1.25 and 0.8) vs. naïve cells for Up and Down, respectively. The pie diagram applies to 94 genes that satisfy the threshold.

**Figure 4 ncrna-08-00072-f004:**
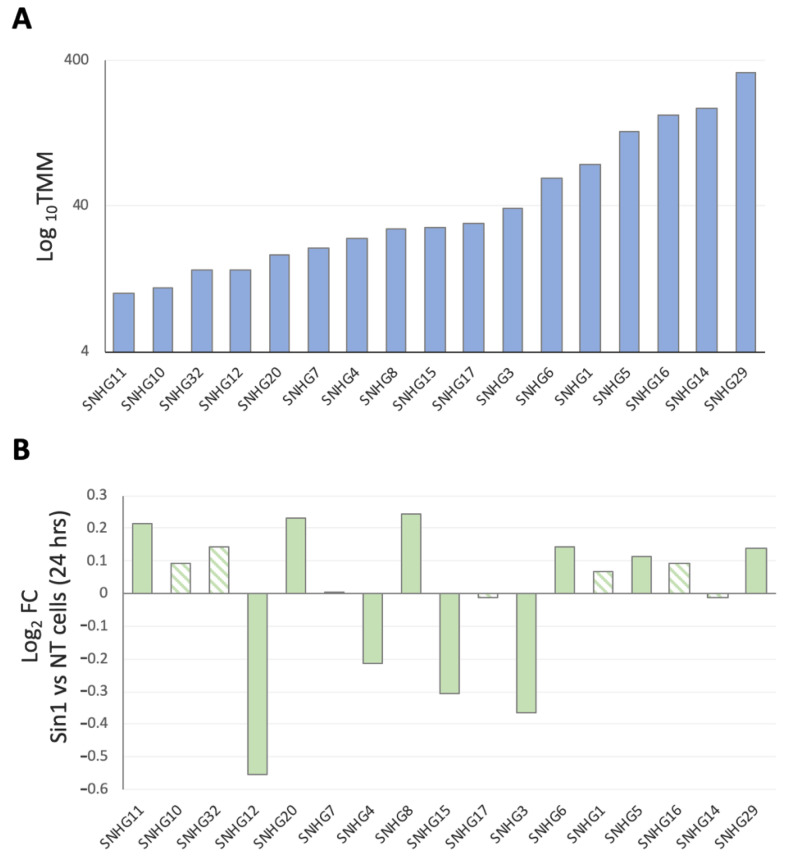
Expression of members of the small nucleolar RNA host genes (SNHGs). (**A**) Mean expressions (by log_10_TMM) of the 17 SNHG members identified by DE lncRNA analysis. (**B**) Fold changes (FCs) following treatment with 50 μM Sin1 are shown for each of the SNHGs. FCs below and above zero show downregulated and upregulation of the indicated genes. Note that the values are logFC. Despite the small effect of the differential expression genes, the filled bar shows statistically significant genes, and the striped bar indicates a gene belonging to the SNHG group, but failed to reach FDR (*p*-value < 0.05).

**Figure 5 ncrna-08-00072-f005:**
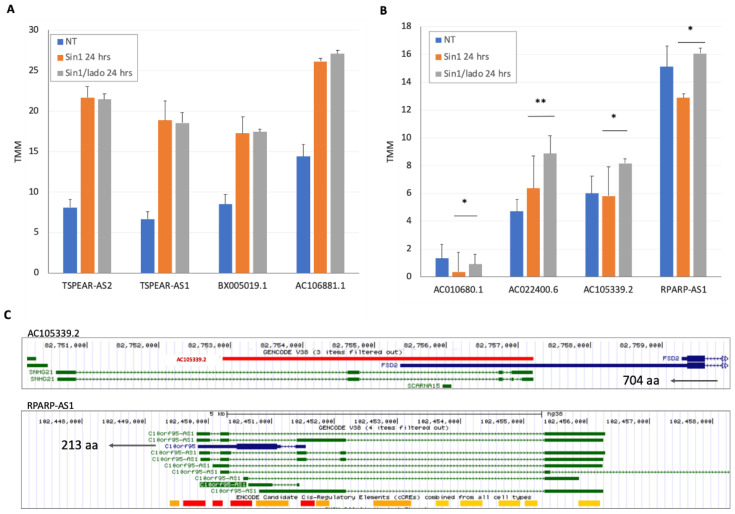
Sample of the lncRNAs by the measured expression levels in naïve untreated cells (NT), cells exposed to Sin1 (24 h) with or without ladostigil. (**A**) The transcripts shown are those with maximal induction by Sin 1 relative to NT. The average TMM and the standard deviations from biological triplicates are shown for each experimental group. (**B**) The transcripts shown are those with a maximal effect by ladostigil in the presence of Sin1. The average TMM and the standard deviations from biological triplicates are shown for each experimental group. The statistical significance with the *p*-value FDR ranging from 0.05–0.005 is depicted by * and <0.005 by **. (**C**) The genomic segments of AC105339.2 and RPARP-AS1 cover 10 and 12k nucleotides, respectively. AC105339.2 and RPARP-AS1 show the overlap between the lncRNA transcript and the gene in the vicinity. AC105339.2 acts as antisense to FSD2 (704 aa) and RPARP-AS1 is the antisense for C10orf95 (213 aa). Numerous transcripts of RPARP-AS1 are shown above the ENCODE annotations for enhancers (red) and proximal promotors (yellow).

**Figure 6 ncrna-08-00072-f006:**
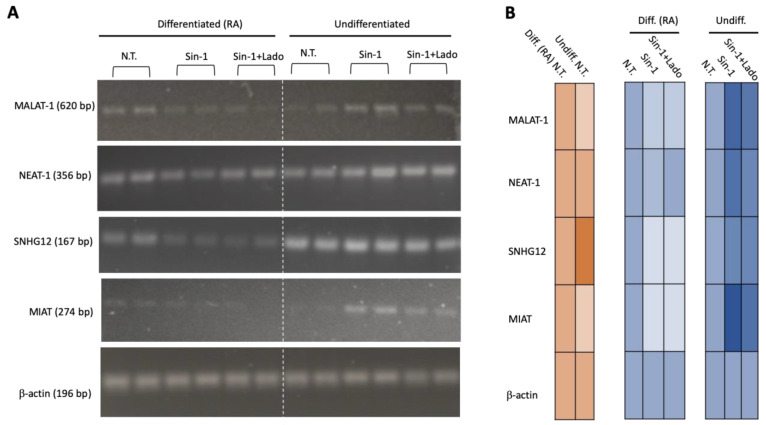
Differentially expressed (DE) ncRNAs following 24 h of exposure to Sin1 relative to untreated cells. (**A**) RT-PCR results on Agarose gel. The RT-PCR product of β-actin was used as an internal control. Each condition is shown for two biological duplicates. The indicated ncRNA results are for the untreated culture (N.T.) and cells in the presence of Sin1, and Sin1 and ladostigil (5.4 μM). (**B**) Quantification of the results in (A) by a heatmap visualization. Orange, comparison of the baseline of the untreated cells. Blue, changes in cells by Sin1, and Sin1 and ladostigil (5.4 μM).

**Table 1 ncrna-08-00072-t001:** Differentially expressed lncRNA genes in cells exposed to Sin1 (24 h).

LncRNA	LncRNA—Full Name	FDR	DE (Fold)	TMM ^a^
MALAT1	Metastasis-associated lung adenocarcinoma transcript 1	2.5 × 10^−4^	1.24	1963.7
HAND2-AS1	HAND2 antisense RNA 1	6.7 × 10^−25^	1.33	671.8
GABPB1-AS1	GABPB1 antisense RNA 1	1.7 × 10^−18^	1.38	280.8
MIAT	Myocardial infarction associated transcript	6.3 × 10^−21^	1.35	176.9
NEAT1	Nuclear paraspeckle assembly transcript 1	3.5 × 10^−5^	1.49	154.0
CASC15	Cancer susceptibility 15	1.5 × 10^−16^	1.27	85.8
AC093849.2	Novel transcript	2.9 × 10^−33^	1.46	85.0
CCDC144NL-AS1	CCDC144NL antisense RNA 1	1.6 × 10^−12^	1.32	67.5
MIR137HG	MIR137 host gene	4.5 × 10^−23^	1.47	58.7
GABPB1-IT1	GABPB1 intronic transcript	4.6 × 10^−15^	1.35	45.6
LINC02268	Long intergenic non-protein coding RNA 2268	1.2 × 10^−16^	1.45	44.6
TTN-AS1	TTN antisense RNA 1	1.2 × 10^−6^	1.26	43.0
LINC01833	Long intergenic non-protein coding RNA 1833	6.8 × 10^−14^	1.32	42.0
LINC00632	Long intergenic non-protein coding RNA 632	5.0 × 10^−8^	1.31	40.2
AL392083.1	Novel transcript, sense intronic to ADARB2	1.8 × 10^−33^	1.78	37.6
GATA3-AS1	GATA3 antisense RNA 1	2.3 × 10^−8^	1.32	36.5
AC012354.1	Novel transcript	2.1 × 10^−8^	1.28	32.7
AC015813.1	Novel transcript	4.3 × 10^−7^	1.35	31.8
AC136621.1	Novel transcript (TEC)	2.6 × 10^−27^	0.56	37.7
AL645608.2	Novel transcript	4.7 × 10^−8^	0.71	35.9
AC010931.1	Novel transcript	1.5 × 10^−19^	0.68	34.9
LINC01128	Long intergenic non-protein coding RNA 1128	1.2 × 10^−16^	0.71	32.2
EP400P1	EP400 pseudogene 1	4.3 × 10^−6^	0.78	32.1

^a^ TMM, trimmed mean of M-values.

**Table 2 ncrna-08-00072-t002:** ncRNA upregulated by ladostigil treatment under oxidative stress conditions.

ncRNA	GH Score	Total Score	Elite GH ^a^	TSS (kb)	# of DB Evidence	TFs	Coding Targets	ncRNA Targets	Selected Target
AC010680.1	0.3	**250.7**	*	64.0	1/5	1.2	2	3	FKBP7
AC022400.6	**1.9**	**250.7**	**	486.4	5/5	4.6	10	13	SEC24C
AC105339.2	**2.1**	11.3	*	24.1	5/5	9.8	7	5	WHAMM
RPARP-AS1	**2.1**	**257.3**	**	546.8	5/5	6.6	13	5	PPRC1

^a^ GeneHancer (GH) definition of elite (*) gene based on the enhancer score and/or high confidence for the gene targets. High confidence scores are shown in bold. Genes that meet high confidence GH and Total scores are marked (**). Selected gene, a representative gene from the set of coding targets.

**Table 3 ncrna-08-00072-t003:** Forward and reverse oligonucleotide primers used for gel-based RT-PCR assays.

Symbol	Transcript	Forward Primer	Reverse Primer	Amplicon (nt)
MALAT1	NR_144568.1	GCTCTGTGGTGTGGGATTGA	CTCGGGCGAGGCGTATTTAT	386
NEAT1	NR_028272.1	GGGACAACATTGACCAACGC	ACCACGGTCCATGAAGCATT	356
MIAT	NR_033321.2	TCCCATTCCCGGAAGCTAGA	GAGGCATGAAATCACCCCCA	274
SNHG12	NR_146383.1	CCTTCTCTCGCTTCGGACTG	ATCTGCTTAAGTACGCCGGG	167
ACTB	NM_001101.5	ACAGAGCCTCGCCTTTGCCGA	CATGCCCACCATCACGCCCTGG	196

## Data Availability

All data used in this study are available as processed data in the Appendix A. The original data from RNA-seq are publicly available from ArrayExpress under accession number E-MTAB-10450.

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
