# Peer review of "Oxidative Stress and Its Modulation by Ladostigil Alter the Expression of Abundant Long Non-Coding RNAs in SH-SY5Y Cells"

_ncrna, 2022, doi:10.3390/ncrna8060072_

Round 1

Reviewer 1 Report (New Reviewer)

The authors may try to perform the mechanistic study on the lncRNA identified in this study. Results are interesting but the manuscript will improve on the functional characterisation of these lncRNAs

Author Response

Comments to Reviewer 1

We agree with the reviewer that adding detailed functional mechanistic experiments could benefit the manuscript. However, we argue that the manuscript is already very dense and addresses numerous aspects of the ncRNA in SH-SY5Y cells, including quite a few of indirect and literature-based functional information.

For example, we inspect the impact of lncRNAs levels in view of cell differentiation for several of the abundant lncRNAs. We show that by exposing cells to prolonged treatment of retinoic acid (RA), cells are shifted where the cells are robust to Sin-1 oxidation and lncRNAs such as MIAT, MALAT-1 SNHG12 are suppressed.

We added an overlooked functional aspect to the observed changes in lncRNA by ladostigil based on the notion of ceRNA regulation. To this end, we added a paragraph (revised, lines 372-390, red). We also included this aspect explicitly in the ‘Discussion’, presenting the exceptional occurrences of miR-106a MBS in many of the abundant lncRNAs discussed in this study (revised: lines 473-482 red, added MBS for SNHGs in revised suppl. Table S4). We added supporting information in Supplementary Table S5 and added supportive references as needed.

Reviewer 2 Report (New Reviewer)

In this manuscript, the authors have predicted the possible role of lncRNA against Sin1-associated negative impact on SH-SY5Y cells. I have the following concerns

 Abstract: Not clearly written. The upper half of the abstract is clear; however, the down one is confusing. I suggest rewriting the abstract.  

Introduction:

line 56: In mice should be in mice;

line 78: What does the meaning of mild stress….Clarify in the introduction;

line 81: ladostigil suggests an overlooked…What is this overlooked contribution?

Material and method

Provide the catalog number of each reagent used in the study.

 Results

The quality and uniformity of all the figures are poor. For instance, Figure 1: 1A has no X-axis line, and 1C and D have no Y-axis line. The font size and style are not the same as the text. I suggest working on all the figures and making the figure correct, and maintaining uniformity.

Lines 175-76: Whether this sentence is required under this heading as the suppression of cell death is discussed.

Line 182: Is it supplementary figure S1 or Figure 1A? I did not find supplemental figure S1

Figure 1: What is the cell death in the normal SH-SY5Y cells? I am surprised to see 8% cell death after H2O2 as 4-6% cell death is expected in normal conditions.

Figure 1B: The author should range the Y axis correctly according to the concentrations (0-350mM) used.

Figure 1 D: What is the difference between Figure 1C and 1D. what precisely the author observed with these two experiments is unclear.

How the author thinks this death is due to oxidative stress. Whether the author have tested the oxidative stress in the cells against the tested concentrations.

Line 208: These results provide additional support…..What the author means by additional support

Figure 2: Figure 2 is not clear. The statement which is written in the text is not matching with the Figure 2-C.

Line 241: Figure 2C shows the top-expressing ncRNAs….What does it mean top-expressing ncRNAs?

Line 287: ‘We discovered’ should be “we found or observed”

Line 308: The concentration of Sin1 was 50 mM or 50 mM?

--Result 3.1: The author should write the results elaborately, such as how much % is reused and all.

Discussion should be more focused, and I suggest discussing the connection between oxidative stress and the misregulated lncRNA.

Author Response

In this manuscript, the authors have predicted the possible role of lncRNA against Sin1-associated negative impact on SH-SY5Y cells. I have the following concerns

Abstract: Not clearly written. The upper half of the abstract is clear; however, the down one is confusing. I suggest rewriting the abstract.  

Reply: Thanks. As asked we rewrote the abstract, making it clear and easier to read. We sharpened the main findings the added short conclusions.

Thanks for careful reading and useful comments that improved the delivery and smooth reading.

Introduction:

line 56: In mice should be in mice;

Corrected

line 78: What does the meaning of mild stress….Clarify in the introduction;

Clarified (lines 204-206, line 219, red)

line 81: ladostigil suggests an overlooked…What is this overlooked contribution?

Rephrased

Material and method

Provide the catalog number of each reagent used in the study.

Added

 Results

The quality and uniformity of all the figures are poor. For instance, Figure 1: 1A has no X-axis line, and 1C and D have no Y-axis line. The font size and style are not the same as the text. I suggest working on all the figures and making the figure correct, and maintaining uniformity.

Thanks for the comment. We reformatted the Figures to add missing axis. Please note that for clarity using fonts that are simple (i.e. Arial) is easier to read. We kept clarity as a main guideline in Figures.

Lines 175-76: Whether this sentence is required under this heading as the suppression of cell death is discussed.

Thanks. The sentence was removed as it was indeed out of place.

Line 182: Is it supplementary figure S1 or Figure 1A? I did not find supplemental figure S1

Sorry, apparently it was not uploaded with the Supplementary materials. It was added in the end of the document.

Figure 1: What is the cell death in the normal SH-SY5Y cells? I am surprised to see 8% cell death after H2O2 as 4-6% cell death is expected in normal conditions.

Actually, the cells are effective in coping with H2O2 (due to high level of catalase). This is one of the reasons to switch to Sin1 that exerts an on-going stress condition. Note that the experiment is a short time (5 hrs) and cell death is not fully established.

Figure 1B: The author should range the Y axis correctly according to the concentrations (0-350mM) used.

This is done as requested. The Y-axis is the % relative to untreated cells and all tested concentrations are indicated.

Figure 1 D: What is the difference between Figure 1C and 1D. what precisely the author observed with these two experiments is unclear. How the author thinks this death is due to oxidative stress. Whether the author have tested the oxidative stress in the cells against the tested concentrations.

Figure 1C and 1D focuses on two aspects of cell death the necrotic PI permeability and the Annexin V staining that is indicative for the apoptotic wave). We added a sentence to clarify the difference in the revised manuscript.

Line 208: These results provide additional support…..What the author means by additional support

Sorry, uncleared writing. Replaced

Figure 2: Figure 2 is not clear. The statement which is written in the text is not matching with the Figure 2-C.

Thanks for noting this mistake. Corrected.

Line 241: Figure 2C shows the top-expressing ncRNAs….What does it mean top-expressing ncRNAs?

Replaced wording to a high-expressing lncRNAs

Line 287: ‘We discovered’ should be “we found or observed”

Indeed, replaced

Line 308: The concentration of Sin1 was 50 mM or 50 mM?

Corrected, it is as in all other places 50 microM

--Result 3.1: The author should write the results elaborately, such as how much % is reused and all.

Corrected when needed. We elaborate on context in some places within the Result section so we will not have to return to it in length in the discussion. It allows the discussion to be more focused (as requested). Note that sources for Fig/ Tables are available in supporting tables (Tables S1-S5).

Discussion should be more focused, and I suggest discussing the connection between oxidative stress and the misregulated lncRNA.

We revised the discussion and added the aspect of ceRNA (per comment from the other reviewer). We elaborated on the of oxidative stress and lncRNA dysregulation. We removed repetitions and improved the writing as suggested.

Round 2

Reviewer 1 Report (New Reviewer)

Manuscript has improved.

Reviewer 2 Report (New Reviewer)

This revised version of the paper responds to all points raised.

This manuscript is a resubmission of an earlier submission. The following is a list of the peer review reports and author responses from that submission.

Round 1

Reviewer 1 Report

In this paper, the authors investigated by RNA-seq experiments the expression of long non- coding RNAs (lncRNAs)s in SH-SY5Y cells treated with oxidative insults.

The authors used SHSY5Y neuroblastoma cells. Teh results obtained are interesting; nevertheless is mandatory to perform selected experiments in  primary cells. In this sense, the authors refers to neuroinflammatio and microglia activation: why did they use SHSY5Y cells?

Methods section: 2x104 cells per well in 96-well plates: it is a very large number of cells for a 96 well plate. Please explain.

Reviewer 2 Report

Dear Authors,

Recently non-coding RNA has become an interesting target in regard to many physological and pathological conditions. Approaches to decipher their role has been increasing our knowledge abot regulatory mechanisms in gene regulation. Research undertaken in the current manuscript by Zohar et al, shows the potential role of non-coding RNAs in regulating oxidative stress response in neuroblastoma cell line. The story seems to be convincing up to the stage of RNA-Seq however then selection and testing of DJ-1 for deeper analysis creates some doubts. Changes in non-coding RNAas are very small after the treatment and even after the prevention of treatment. Not only small number of genes is affected but also the fold change is very small. However authors still claim this is a very important mechanism. Selection of DJ-1/PARK7 for testing as regulator of NRF2 has no explenation, is not based on RNA-Seq or any other reuslts provided in the manuscript. It seems to be separate story. There are many flows in the text. Discussion style is mixed with results. Non of the transcripts was tested and confirmed from RNA-Seq by any other methods. There is no connection between inititaly described results including RNA-Seq and final conclusions especailly related to DJ-1. Dj-1 occures suddenly without any background, and is linked by authors with NRF2 and antioxidative system. All conclusion about Dj-1 is based only on one qPCR and agarose gel with separated bands. I belive that this work is not suitbale for publication and I would say to reject this manuscript.

Abstract need some additional polishing, it provides most of the findings from the paper but some sentences are loosely conected such as statments about enhancers line 21 or NRF2 in the line 23. Please try to connect those findings to create more flowless story that is shown in your article.

Line 33, cellular homeostastis is not a „hallmark”. This is very true for each and every system in the body. This is just general sentence that should be changed.

Line 41, „signified” it is not the best world here. It is not totally clear what authors ment by this word.

Line 50, „such as and” – and is rather not necessary here

Line 55-59, all this responses seems to be true in regard to oxidative stress, but their significance is neither well described or introduced. Description of autophagy process or rather its function comes suddenly without really connection with the rest of pragraph. Please try to improve this by flowless connection between different ideas. Why sth is activated or inhibited, why is needed or not needed.

Line 60, There is „asseyed” and should not be assesed? Furhter in the same line, „induced by extreme conditions”? This is exagerated formulation, it is much better if you simply describe what you did and how, what was insult. Extreme condtion as protein misfolding? It should be changed.

Line 63-64, you indicate that Sin1 creates continuse oxidative stress so the next part of sentence saying „prolonged oxidative stress” is marely a repetition.

Line 66-68, Despite authors claim that non-coding RNAs are so important there is no single word in introduction about those RNAs. Introduction is very poor in this regard.

Line 66-67, the sentence is meaningless. Authors already say they want to study oxidative stress and non-coding RNAs so no need to say the same just rephrasing the sentence.

Line 109. the sentence itself about increased fluorescence is rathere not a significant in the methodology. In general those are standard methods used for studying cell death, so either it should be only mentioned, to study apoprotosis annexin and PI were used. Without too many basic details.

Line 119-120, why there is information about spinning? Was it left accidently, or ther rest was removed? If this is according to protocol should not be mentioned.

Line 143, cutoff of log2FC from 0.3 is a very small difference, just about 20% reduction or upregulation.

Line 158, Figure 1. Is the difference in number of viable cells indeed statistical. It is true that changes is visible but is this a significant change?

Line 174-176, description for how is staining of annexin done is not a place for results. This is general information, from the kit description and could be omitted. Additionaly why here authors do test for annexin, apoptosis and previous assay was determined as detection of necrosis only?

Line 177, based on simple viabaility test authors can not propose anything. Sentence about changes in gene expression is not a result should be removed.

Fig1 and Fig2b. Why calculations and bars were done only in the second experiment?

Line 187’ please be more specific and adequate what authors mean by „untreated”? Beacuse it is written (untreated, incubated with 5.4 uM ladostigil), The line 195 is a continuation of that question

Line 195. „In the presence of ladostigil (5.4 mM), a 2-fold reduction in the fraction of dead cells relative to the Sin1 baseline”. To which graph this statment corresponds. There is no number that can be considered as two fold change. In Fig 2C authors have basal level of 6.5 than 10.9 and 8.7%? Something seems incorrect. And description as well approach seems to be confusing. Why at once control is no treatment and the other time as in Fig 2C the control is no treatment + lado? It doesnt’t seem to have any statistical effect. Additionaly when it is combined with further results from RNASeq where 94 ncRNAs are satisfing statistical differences but with only very small changes.

Line 219-220, results and discussion are seprate paragraphs, thereofre please try not to mix those parts. There is only suggestion which is rather discussion part

Line 210 and Figure 3b, what are mito – transcripts. Is this indeed a class or based on origin. LncRNAs or unprocessed transcipts are class of ncRNAs but mitochodnrial transcripts? What kind of class is this. This seems to be confused with other way of seprataion that authors tried to do which is functional subgroup? It is not clear.

Line 213 – where this transcripts come from MT-RNR1 and MT-RNR2? It can’t be found in the S1 table. Indeed there are several highly expressed other mitochodnrial genes, however statment about very important role is not supproted at this point by other observations

Line 231 „In addition, XIST, FTX, and JPX are ncRNAs that govern the X-chromosomal inhibition process in the female genome, supporting the role of abundant ncRNAs in chromatin compaction”. What is the sense of this sentence. How it came here after first talking about mitochondria? There seems to be no rationale for this result here? How and why female genes are important, are those cells of female origin? Is there any reason why authors wanted to foucs on those genes?

Line 238-246 this is not results but discussion.

Line 256, „1.2e-02” this pvalue is so smal lvalue should be used as 0.012

Line 275 „1.25 and 0.8” as a cutoff, while previously authors were using the log values. Please be consistent. Suppl tables are as log fold as well, while in the text authors use normal FC values

Table 1 shows DE genes which are upregulated after the treatment with Sin1 less than 1.5 fold.

Line 278-286, there is no result it is discussion related.

Line 293-298 discussion, not a result

Paragraph 3.6 is a big part mixture with discussion style. Should be changed.

Fig7a and 7b ics very chaotic, hard to read. Should much better described, and organized. Which line of the gel is what. Which band correspond to the which amplicon. Moroeveor, how DJ-1/PARK7 was selected. There is no eveidence from RNA-Seq data, and authors jump into NRF2 paragraph studing specific mechanism in which DJ-1 is a key player. Most of data presented till this part seems to be loosely connected, not making a great and hard evidnce for the presented conclusions. Why H202 + lado on Fig 7a is lacking upper bands. Therefore story with DJ-1 seems to be not related with RNA-Seq.

Fig 402-414 is not result

In general very often results are accompanied by many senetences being a discussion and reference to other articles.

Reviewer 3 Report

In the manuscript by Zohar et al. investigated the effect of oxidative stress on transcriptional profiles of long noncoding RNAs (lncRNAs) in SHSY5Y cells. This is an interesting study showing that oxidative stress affected the lncRNA transcriptional profiles in neuronal cells and identified several candidate genes whose expressions were altered in relation to the changes of lncRNAs transcriptions induced by oxidative stress. I found several concerns before publication.

Figure 1, the authors showed that ladostigil rescued H2O2 cytotoxicity, but performed subsequent experiments by using Sin-1 and ladostigil. I understand that the authors focused on "prolonged and steady" oxidative stress, but I do not understand the significance of Figure 1.

Figure 2B needs statistical analysis. It is not clear that the increase of 6.5% to 10.9% in dead cells was actually significant. Please add more explanation to the shift from apoptosis to driving cell death in Sin-1-treated SHSY5Y cells.

Regarding the "3.5 Cancer-related SNHG family members...", because SHSY5Y cells are neuroblastoma cells, it is not clear these cancer-related genes are actually relate to neuronal damage.

Figure 7, particularly Figure 7C, is not supported by the data presented here. DJ-1 and Nrf-2 are clearly involved in the rescue of cells from oxidative stress, but the current data are too preliminary to include Figure 7C as a summary.

In the discussion, lines450-452 need a ref. Lines 474-513 can be shortened because these parts are not supported by the data.

Author Response

Please see attachment (reply)

Round 2

Reviewer 1 Report

The authors Sin1 in living cells acts as a peroxynitrite donor 
(ONOO−) and generates nitric oxide (NO) and superoxide (O2−). The pro-oxidant effect in cells used in the present paper should be clearly demonstrated, besides MTT and flow cytometry.

Reviewer 2 Report

Dear Authors,

Provided responses and changes to he text improved the quality of the manuscript however at the same time showed that this work is very initial.  Unfortuantely included changes do not change my general perception of the work. Sequencing as a main result is in my opinion not enough to aspire for publication in Antioxidants as a orignial research. It has no validation by PCR, no mechanisms are provided. According to me this is too basic and too initial work to drive more significant conclusions. The bioinformatic part such as data analysis seems to be expanded. However most of the conclusions are just predictions based on the possible function of those genes especially in the context of very small fold changes. This is interesting work, however at this point with no aspiration to be published in Antioxidants. With the best intentions, I would advice to do additional work, verify changes in the expression by other methods, test some targets, possible patways, add some mechanistic appoaches in order to aspire for higher quality work. At this stage I would say that RNASeq is merely a first step with much more work needed. Therefore I do not recomend this work for publication in Antioxidants.